# Tight Continuous Relaxation of the Balanced $k$-Cut Problem

**Syama Sundar Rangapuram, Pramod Kaushik Mudrakarta and Matthias Hein**
Department of Mathematics and Computer Science
Saarland University, Saarbrücken

## Abstract

Spectral Clustering as a relaxation of the normalized/ratio cut has become one of the standard graph-based clustering methods. Existing methods for the computation of multiple clusters, corresponding to a balanced $k$-cut of the graph, are either based on greedy techniques or heuristics which have weak connection to the original motivation of minimizing the normalized cut. In this paper we propose a new tight continuous relaxation for any balanced $k$-cut problem and show that a related recently proposed relaxation is in most cases loose leading to poor performance in practice. For the optimization of our tight continuous relaxation we propose a new algorithm for the difficult sum-of-ratios minimization problem which achieves monotonic descent. Extensive comparisons show that our method outperforms all existing approaches for ratio cut and other balanced $k$-cut criteria.

## 1   Introduction

Graph-based techniques for clustering have become very popular in machine learning as they allow for an easy integration of pairwise relationships in data. The problem of finding $k$ clusters in a graph can be formulated as a balanced $k$-cut problem [1, 2, 3, 4], where ratio and normalized cut are famous instances of balanced graph cut criteria employed for clustering, community detection and image segmentation. The balanced $k$-cut problem is known to be NP-hard [4] and thus in practice relaxations [4, 5] or greedy approaches [6] are used for finding the optimal multi-cut. The most famous approach is spectral clustering [7], which corresponds to the spectral relaxation of the ratio/normalized cut and uses $k$-means in the embedding of the vertices found by the first $k$ eigenvectors of the graph Laplacian in order to obtain the clustering. However, the spectral relaxation has been shown to be loose for $k = 2$ [8] and for $k > 2$ no guarantees are known of the quality of the obtained $k$-cut with respect to the optimal one. Moreover, in practice even greedy approaches [6] frequently outperform spectral clustering.

This paper is motivated by another line of recent work [9, 10, 11, 12] where it has been shown that an exact continuous relaxation for the two cluster case ($k = 2$) is possible for a quite general class of balancing functions. Moreover, efficient algorithms for its optimization have been proposed which produce much better cuts than the standard spectral relaxation. However, the multi-cut problem has still to be solved via the greedy recursive splitting technique.

Inspired by the recent approach in [13], in this paper we tackle directly the general balanced $k$-cut problem based on a new tight continuous relaxation. We show that the relaxation for the asymmetric ratio Cheeger cut proposed recently by [13] is loose when the data does not contain $k$ well-separated clusters and thus leads to poor performance in practice. Similar to [13] we can also integrate label information leading to a transductive clustering formulation. Moreover, we propose an efficient algorithm for the minimization of our continuous relaxation for which we can prove monotonic descent. This is in contrast to the algorithm proposed in [13] for which no such guarantee holds. In extensive experiments we show that our method outperforms all existing methods in terms of the

achieved balanced $k$-cuts. Moreover, our clustering error is competitive with respect to several other clustering techniques based on balanced $k$-cuts and recently proposed approaches based on non-negative matrix factorization. Also we observe that already with small amount of label information the clustering error improves significantly.

## 2   Balanced Graph Cuts

Graphs are used in machine learning typically as similarity graphs, that is the weight of an edge between two instances encodes their similarity. Given such a similarity graph of the instances, the clustering problem into $k$ sets can be transformed into a graph partitioning problem, where the goal is to construct a partition of the graph into $k$ sets such that the cut, that is the sum of weights of the edge from each set to all other sets, is small and all sets in the partition are roughly of equal size.

Before we introduce balanced graph cuts, we briefly fix the setting and notation. Let $G(V, W)$ denote an undirected, weighted graph with vertex set $V$ with $n = |V|$ vertices and weight matrix $W \in \mathbb{R}_+^{n \times n}$ with $W = W^T$. There is an edge between two vertices $i, j \in V$ if $w_{ij} > 0$. The cut between two sets $A, B \subset V$ is defined as $\mathrm{cut}(A, B) = \sum_{i \in A, j \in B} w_{ij}$ and we write $\mathbf{1}_A$ for the indicator vector of set $A \subset V$. A collection of $k$ sets $(C_1, \ldots, C_k)$ is a partition of $V$ if $\cup_{i=1}^k C_i = V$, $C_i \cap C_j = \emptyset$ if $i \neq j$ and $|C_i| \geq 1$, $i = 1, \ldots, k$. We denote the set of all $k$-partitions of $V$ by $P_k$. Furthermore, we denote by $\Delta_k$ the simplex $\{x : x \in \mathbb{R}^k, \ x \geq 0, \ \sum_{i=1}^k x_i = 1\}$.

Finally, a set function $\hat{S} : 2^V \to \mathbb{R}$ is called submodular if for all $A, B \subset V$, $\hat{S}(A \cup B) + \hat{S}(A \cap B) \leq \hat{S}(A) + \hat{S}(B)$. Furthermore, we need the concept of the Lovasz extension of a set function.

**Definition 1** *Let $\hat{S} : 2^V \to \mathbb{R}$ be a set function with $\hat{S}(\emptyset) = 0$. Let $f \in \mathbb{R}^V$ be ordered in increasing order $f_1 \leq f_2 \leq \ldots \leq f_n$ and define $C_i = \{j \in V \mid f_j > f_i\}$ where $C_0 = V$. Then $S : \mathbb{R}^V \to \mathbb{R}$ given by, $S(f) = \sum_{i=1}^n f_i \big( \hat{S}(C_{i-1}) - \hat{S}(C_i) \big)$, is called the **Lovasz extension** of $\hat{S}$. Note that $S(\mathbf{1}_A) = \hat{S}(A)$ for all $A \subset V$.*

The Lovasz extension of a set function is convex if and only if the set function is submodular [14]. The cut function $\mathrm{cut}(C, \overline{C})$, where $\overline{C} = V \backslash C$, is submodular and its Lovasz extension is given by $\mathrm{TV}(f) = \frac{1}{2} \sum_{i,j=1}^n w_{ij} |f_i - f_j|$.

### 2.1   Balanced $k$-cuts

The balanced $k$-cut problem is defined as

$$\min_{(C_1,\ldots,C_k) \in P_k} \sum_{i=1}^k \frac{\mathrm{cut}(C_i, \overline{C_i})}{\hat{S}(C_i)} =: \mathrm{BCut}(C_1, \ldots, C_k) \tag{1}$$

where $\hat{S} : 2^V \to \mathbb{R}_+$ is a balancing function with the goal that all sets $C_i$ are of the same "size". In this paper, we assume that $\hat{S}(\emptyset) = 0$ and for any $C \subsetneq V$, $C \neq \emptyset$, $\hat{S}(C) \geq m$, for some $m > 0$. In the literature one finds mainly the following submodular balancing functions (in brackets is the name of the overall balanced graph cut criterion $\mathrm{BCut}(C_1, \ldots, C_k)$),

$$\hat{S}(C) = |C|, \qquad\qquad\qquad\qquad\qquad \text{(Ratio Cut)}, \tag{2}$$
$$\hat{S}(C) = \min\{|C|, |\overline{C}|\}, \qquad\qquad\qquad \text{(Ratio Cheeger Cut)},$$
$$\hat{S}(C) = \min\{(k-1)|C|, \overline{C}\} \qquad\qquad \text{(Asymmetric Ratio Cheeger Cut)}.$$

The *Ratio Cut* is well studied in the literature e.g. [3, 7, 6] and corresponds to a balancing function without bias towards a particular size of the sets, whereas the *Asymmetric Ratio Cheeger Cut* recently proposed in [13] has a bias towards sets of size $\frac{|V|}{k}$ ($\hat{S}(C)$ attains its maximum at this point) which makes perfect sense if one expects clusters which have roughly equal size. An intermediate version between the two is the *Ratio Cheeger Cut* which has a symmetric balancing function and strongly penalizes overly large clusters. For the ease of presentation we restrict ourselves to these balancing functions. However, we can also handle the corresponding weighted cases e.g., $\hat{S}(C) = \mathrm{vol}(C) = \sum_{i \in C} d_i$, where $d_i = \sum_{j=1}^n w_{ij}$, leading to the *normalized cut*[4].

# 3 Tight Continuous Relaxation for the Balanced $k$-Cut Problem

In this section we discuss our proposed relaxation for the balanced $k$-cut problem (1). It turns out that a crucial question towards a tight multi-cut relaxation is the choice of the constraints so that the continuous problem also yields a partition (together with a suitable rounding scheme). The motivation for our relaxation is taken from the recent work of [9, 10, 11], where exact relaxations are shown for the case $k = 2$. Basically, they replace the ratio of set functions with the ratio of the corresponding Lovasz extensions. We use the same idea for the objective of our continuous relaxation of the $k$-cut problem (1) which is given as

$$\min_{\substack{F=(F_1,\ldots,F_k),\\ F\in\mathbb{R}_+^{n\times k}}} \sum_{l=1}^{k} \frac{\mathrm{TV}(F_l)}{S(F_l)} \tag{3}$$

$$\begin{aligned}
\text{subject to}: \ &F_{(i)} \in \Delta_k, &&i=1,\ldots,n, &&\text{(simplex constraints)}\\
&\max\{F_{(i)}\} = 1, &&\forall i \in I, &&\text{(membership constraints)}\\
&S(F_l) \geq m, &&l=1,\ldots,k, &&\text{(size constraints)}
\end{aligned}$$

where $S$ is the Lovasz extension of the set function $\hat{S}$ and $m = \min_{C \subsetneq V,\ C\neq\emptyset} \hat{S}(C)$. We have $m = 1$, for *Ratio Cut* and *Ratio Cheeger Cut* whereas $m = k - 1$ for *Asymmetric Ratio Cheeger Cut*. Note that TV is the Lovasz extension of the cut functional $\mathrm{cut}(C, \overline{C})$. In order to simplify notation we denote for a matrix $F \in \mathbb{R}^{n\times k}$ by $F_l$ the $l$-th column of $F$ and by $F_{(i)}$ the $i$-th row of $F$. Note that the rows of $F$ correspond to the vertices of the graph and the $j$-th column of $F$ corresponds to the set $C_j$ of the desired partition. The set $I \subset V$ in the membership constraints is chosen adaptively by our method during the sequential optimization described in Section 4.

An obvious question is how to get from the continuous solution $F^*$ of (3) to a partition $(C_1, \ldots, C_k) \in P_k$ which is typically called *rounding*. Given $F^*$ we construct the sets, by assigning each vertex $i$ to the column where the $i$-th row attains its maximum. Formally,

$$C_i = \{j \in V \mid i = \arg\max_{s=1,\ldots,k} F_{js}\}, \quad i=1,\ldots,k, \quad \text{(Rounding)} \tag{4}$$

where ties are broken randomly. If there exists a row such that the rounding is not unique, we say that the solution is weakly degenerated. If furthermore the resulting set $(C_1, \ldots, C_k)$ do not form a partition, that is one of the sets is empty, then we say that the solution is strongly degenerated.

First, we connect our relaxation to the previous work of [11] for the case $k = 2$. Indeed for symmetric balancing function such as the *Ratio Cheeger Cut*, our continuous relaxation (3) is exact even without membership and size constraints.

**Theorem 1** *Let $\hat{S}$ be a non-negative symmetric balancing function, $\hat{S}(C) = \hat{S}(\overline{C})$, and denote by $p^*$ the optimal value of (3) without membership and size constraints for $k = 2$. Then it holds*

$$p^* = \min_{(C_1,C_2)\in P_2} \sum_{i=1}^{2} \frac{\mathrm{cut}(C_i, \overline{C_i})}{\hat{S}(C_i)}.$$

*Furthermore there exists a solution $F^*$ of (3) such that $F^* = [\mathbf{1}_{C^*}, \mathbf{1}_{\overline{C^*}}]$, where $(C^*, \overline{C^*})$ is the optimal balanced 2-cut partition.*

Note that rounding trivially yields a solution in the setting of the previous theorem.

A second result shows that indeed our proposed optimization problem (3) is a relaxation of the balanced $k$-cut problem (1). Furthermore, the relaxation is exact if $I = V$.

**Proposition 1** *The continuous problem (3) is a relaxation of the $k$-cut problem (1). The relaxation is exact, i.e., both problems are equivalent, if $I = V$.*

The row-wise simplex and membership constraints enforce that each vertex in $I$ belongs to exactly one component. Note that these constraints alone (even if $I = V$) can still not guarantee that $F$ corresponds to a $k$-way partition since an entire column of $F$ can be zero. This is avoided by the column-wise size constraints that enforce that each component has at least one vertex.

If $I = V$ it is immediate from the proof that problem (3) is no longer a continuous problem as the feasible set are only indicator matrices of partitions. In this case rounding yields trivially a partition. On the other hand, if $I = \emptyset$ (i.e., no membership constraints), and $k > 2$ it is not guaranteed that rounding of the solution of the continuous problem yields a partition. Indeed, we will see in the following that for symmetric balancing functions one can, under these conditions, show that the solution is always strongly degenerated and rounding does not yield a partition (see Theorem 2). Thus we observe that the index set $I$ controls the degree to which the partition constraint is enforced. The idea behind our suggested relaxation is that it is well known in image processing that minimizing the total variation yields piecewise constant solutions (in fact this follows from seeing the total variation as Lovasz extension of the cut). Thus if $|I|$ is sufficiently large, the vertices where the values are fixed to 0 or 1 propagate this to their neighboring vertices and finally to the whole graph. We discuss the choice of $I$ in more detail in Section 4.

**Simplex constraints alone are not sufficient to yield a partition:** Our approach has been inspired by [13] who proposed the following continuous relaxation for the *Asymmetric Ratio Cheeger Cut*

$$\min_{\substack{F=(F_1,\ldots,F_k), \\ F\in\mathbb{R}_+^{n\times k}}} \sum_{l=1}^{k} \frac{\mathrm{TV}(F_l)}{\left\|F_l - \mathrm{quant}_{k-1}(F_l)\right\|_1} \tag{5}$$

$$\text{subject to}: F_{(i)} \in \Delta_k, \quad i = 1,\ldots,n, \quad \text{(simplex constraints)}$$

where $S(f) = \left\|f - \mathrm{quant}_{k-1}(f)\right\|_1$ is the Lovasz extension of $\hat{S}(C) = \min\{(k-1)|C|, \overline{C}\}$ and $\mathrm{quant}_{k-1}(f)$ is the $k-1$-quantile of $f \in \mathbb{R}^n$. Note that in their approach no membership constraints and size constraints are present.

We now show that the usage of simplex constraints in the optimization problem (3) is not sufficient to guarantee that the solution $F^*$ can be rounded to a partition for any symmetric balancing function in (1). For asymmetric balancing functions as employed for the *Asymmetric Ratio Cheeger Cut* by [13] in their relaxation (5) we can prove such a strong result only in the case where the graph is disconnected. However, note that if the number of components of the graph is less than the number of desired clusters $k$, the multi-cut problem is still non-trivial.

**Theorem 2** *Let $\hat{S}(C)$ be any non-negative symmetric balancing function. Then the continuous relaxation*

$$\min_{\substack{F=(F_1,\ldots,F_k), \\ F\in\mathbb{R}_+^{n\times k}}} \sum_{l=1}^{k} \frac{\mathrm{TV}(F_l)}{S(F_l)} \tag{6}$$

$$\text{subject to}: F_{(i)} \in \Delta_k, \quad i = 1,\ldots,n, \quad \text{(simplex constraints)}$$

*of the balanced $k$-cut problem* (1) *is void in the sense that the optimal solution $F^*$ of the continuous problem can be constructed from the optimal solution of the 2-cut problem and $F^*$ cannot be rounded into a $k$-way partition, see* (4)*. If the graph is disconnected, then the same holds also for any non-negative asymmetric balancing function.*

The proof of Theorem 2 shows additionally that for any balancing function if the graph is disconnected, the solution of the continuous relaxation (6) is always zero, while clearly the solution of the balanced $k$-cut problem need not be zero. This shows that the relaxation can be arbitrarily bad in this case. In fact the relaxation for the asymmetric case can even fail if the graph is not disconnected but there exists a cut of the graph which is very small as the following corollary indicates.

**Corollary 1** *Let $\hat{S}$ be an asymmetric balancing function and $C^* = \arg\min_{C\subset V} \frac{\mathrm{cut}(C,\overline{C})}{\hat{S}(C)}$ and suppose that $\phi^* := (k-1)\frac{\mathrm{cut}(C^*,\overline{C^*})}{\hat{S}(C^*)} + \frac{\mathrm{cut}(C^*,\overline{C^*})}{\hat{S}(\overline{C^*})} < \min_{(C_1,\ldots,C_k)\in P_k} \sum_{i=1}^{k} \frac{\mathrm{cut}(C_i,\overline{C_i})}{\hat{S}(C_i)}$. Then there exists a feasible $F$ with $F_1 = \mathbf{1}_{\overline{C^*}}$ and $F_l = \alpha_l \mathbf{1}_{C^*}$, $l = 2,\ldots,k$ such that $\sum_{l=2}^{k} \alpha_l = 1, \alpha_l > 0$ for* (6) *which has objective $\sum_{i=1}^{k} \frac{\mathrm{TV}(F_i)}{S(F_i)} = \phi^*$ and which cannot be rounded to a $k$-way partition.*

Theorem 2 shows that the membership and size constraints which we have introduced in our relaxation (3) are essential to obtain a partition for symmetric balancing functions. For the asymmetric

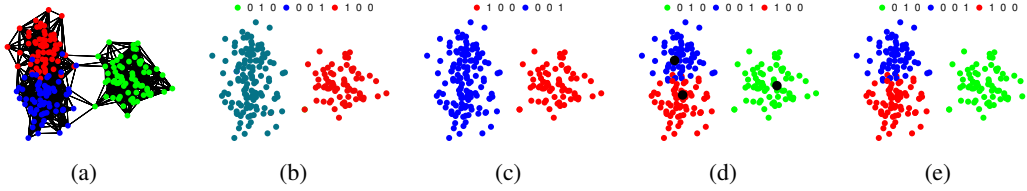

Figure 1: Toy example illustrating that the relaxation of [13] converges to a degenerate solution when applied to a graph with dominating 2-cut. (a) 10NN-graph generated from three Gaussians in 10 dimensions (b) continuous solution of (5) from [13] for $k = 3$, (c) rounding of the continuous solution of [13] does not yield a 3-partition (d) continuous solution found by our method together with the vertices $i \in I$ (black) where the membership constraint is enforced. Our continuous solution corresponds already to a partition. (e) clustering found by rounding of our continuous solution (trivial as we have converged to a partition). In (b)-(e), we color data point $i$ according to $F_{(i)} \in \mathbb{R}^3$.

balancing function failure of the relaxation (6) and thus also of the relaxation (5) of [13] is only guaranteed for disconnected graphs. However, Corollary 1 indicates that degenerated solutions should also be a problem when the graph is still connected but there exists a dominating cut. We illustrate this with a toy example in Figure 1 where the algorithm of [13] for solving (5) fails as it converges exactly to the solution predicted by Corollary 1 and thus only produces a 2-partition instead of the desired 3-partition. The algorithm for our relaxation enforcing membership constraints converges to a continuous solution which is in fact a partition matrix so that no rounding is necessary.

## 4 Monotonic Descent Method for Minimization of a Sum of Ratios

Apart from the new relaxation another key contribution of this paper is the derivation of an algorithm which yields a sequence of feasible points for the difficult non-convex problem (3) and reduces monotonically the corresponding objective. We would like to note that the algorithm proposed by [13] for (5) does not yield monotonic descent. In fact it is unclear what the derived guarantee for the algorithm in [13] implies for the generated sequence. Moreover, our algorithm works for any non-negative submodular balancing function.

The key insight in order to derive a monotonic descent method for solving the sum-of-ratio minimization problem (3) is to eliminate the ratio by introducing a new set of variables $\beta = (\beta_1, \ldots, \beta_k)$.

$$\min_{\substack{F=(F_1,\ldots,F_k), \\ F \in \mathbb{R}_+^{n \times k}, \; \beta \in \mathbb{R}_+^k}} \sum_{l=1}^{k} \beta_l \tag{7}$$

$$\begin{aligned} \text{subject to}: \;\; & \text{TV}(F_l) \leq \beta_l S(F_l), && l = 1, \ldots, k, && \text{(descent constraints)} \\ & F_{(i)} \in \Delta_k, && i = 1, \ldots, n, && \text{(simplex constraints)} \\ & \max\{F_{(i)}\} = 1, && \forall i \in I, && \text{(membership constraints)} \\ & S(F_l) \geq m, && l = 1, \ldots, k. && \text{(size constraints)} \end{aligned}$$

Note that for the optimal solution $(F^*, \beta^*)$ of this problem it holds $\text{TV}(F_l^*) = \beta_l^* S(F_l^*), l = 1, \ldots, k$ (otherwise one can decrease $\beta_l^*$ and hence the objective) and thus equivalence holds. This is still a non-convex problem as the descent, membership and size constraints are non-convex. Our algorithm proceeds now in a sequential manner. At each iterate we do a convex inner approximation of the constraint set, that is the convex approximation is a subset of the non-convex constraint set, based on the current iterate $(F^t, \beta^t)$. Then we optimize the resulting convex optimization problem and repeat the process. In this way we get a sequence of feasible points for the original problem (7) for which we will prove monotonic descent in the sum-of-ratios.

**Convex approximation:** As $\hat{S}$ is submodular, $S$ is convex. Let $s_l^t \in \partial S(F_l^t)$ be an element of the sub-differential of $S$ at the current iterate $F_l^t$. We have by Prop. 3.2 in [14], $(s_l^t)_{j_i} = \hat{S}(C_{l_{i-1}}) - \hat{S}(C_{l_i})$, where $j_i$ is the $i^{th}$ smallest component of $F_l^t$ and $C_{l_i} = \{j \in V \mid (F_l^t)_j > (F_l^t)_i\}$. Moreover, using the definition of subgradient, we have $S(F_l) \geq S(F_l^t) + \langle s_l^t, F_l - F_l^t \rangle = \langle s_l^t, F_l \rangle$.

For the descent constraints, let $\lambda_l^t = \frac{\mathrm{TV}(F_l^t)}{S(F_l^t)}$ and introduce new variables $\delta_l = \beta_l - \lambda_l^t$ that capture the amount of change in each ratio. We further decompose $\delta_l$ as $\delta_l = \delta_l^+ - \delta_l^-$, $\delta_l^+ \geq 0$, $\delta_l^- \geq 0$. Let $M = \max_{f \in [0,1]^n} S(f) = \max_{C \subset V} \hat{S}(C)$, then for $S(F_l) \geq m$,

$$\mathrm{TV}(F_l) - \beta_l S(F_l) \leq \mathrm{TV}(F_l) - \lambda_l^t \left\langle s_l^t, F_l \right\rangle - \delta_l^+ S(F_l) + \delta_l^- S(F_l)$$
$$\leq \mathrm{TV}(F_l) - \lambda_l^t \left\langle s_l^t, F_l \right\rangle - \delta_l^+ m + \delta_l^- M$$

Finally, note that because of the simplex constraints, the membership constraints can be rewritten as $\max\{F_{(i)}\} \geq 1$. Let $i \in I$ and define $j_i := \arg\max_j F_{ij}^t$ (ties are broken randomly). Then the membership constraints can be relaxed as follows: $0 \geq 1 - \max\{F_{(i)}\} \geq 1 - F_{ij_i} \implies F_{ij_i} \geq 1$. As $F_{ij} \leq 1$ we get $F_{ij_i} = 1$. Thus the convex approximation of the membership constraints fixes the assignment of the $i$-th point to a cluster and thus can be interpreted as "label constraint". However, unlike the transductive setting, the labels for the vertices in $I$ are automatically chosen by our method. The actual choice of the set $I$ will be discussed in Section 4.1. We use the notation $L = \{(i, j_i) \mid i \in I\}$ for the label set generated from $I$ (note that $L$ is fixed once $I$ is fixed).

**Descent algorithm:** Our descent algorithm for minimizing (7) solves at each iteration $t$ the following convex optimization problem (8).

$$\min_{\substack{F \in \mathbb{R}_+^{n \times k}, \\ \delta^+ \in \mathbb{R}_+^k, \, \delta^- \in \mathbb{R}_+^k}} \quad \sum_{l=1}^k \delta_l^+ - \delta_l^- \tag{8}$$

$$\begin{aligned}
\text{subject to} : \mathrm{TV}(F_l) &\leq \lambda_l^t \left\langle s_l^t, F_l \right\rangle + \delta_l^+ m - \delta_l^- M, & l &= 1, \ldots k, & &\text{(descent constraints)} \\
F_{(i)} &\in \Delta_k, & i &= 1, \ldots, n, & &\text{(simplex constraints)} \\
F_{ij_i} &= 1, & \forall (i, j_i) &\in L, & &\text{(label constraints)} \\
\left\langle s_l^t, F_l^t \right\rangle &\geq m, & l &= 1, \ldots, k. & &\text{(size constraints)}
\end{aligned}$$

As its solution $F^{t+1}$ is feasible for (3) we update $\lambda_l^{t+1} = \frac{\mathrm{TV}(F_l^{t+1})}{S(F_l^{t+1})}$ and $s_l^{t+1} \in \partial S(F_l^{t+1})$, $l = 1, \ldots, k$ and repeat the process until the sequence terminates, that is no further descent is possible as the following theorem states, or the relative descent in $\sum_{l=1}^k \lambda_l^t$ is smaller than a predefined $\epsilon$. The following Theorem 3 shows the monotonic descent property of our algorithm.

**Theorem 3** *The sequence $\{F^t\}$ produced by the above algorithm satisfies $\sum_{l=1}^k \frac{\mathrm{TV}(F_l^{t+1})}{S(F_l^{t+1})} < \sum_{l=1}^k \frac{\mathrm{TV}(F_l^t)}{S(F_l^t)}$ for all $t \geq 0$ or the algorithm terminates.*

The inner problem (8) is convex, but contains the non-smooth term TV in the constraints. We eliminate the non-smoothness by introducing additional variables and derive an equivalent linear programming (LP) formulation. We solve this LP via the PDHG algorithm [15, 16]. The LP and the exact iterates can be found in the supplementary material.

## 4.1 Choice of membership constraints $I$

The overall algorithm scheme for solving the problem (1) is given in the supplementary material. For the membership constraints we start initially with $I^0 = \emptyset$ and sequentially solve the inner problem (8). From its solution $F^{t+1}$ we construct a $P_k' = (C_1, \ldots, C_k)$ via rounding, see (4). We repeat this process until we either do not improve the resulting balanced $k$-cut or $P_k'$ is not a partition. In this case we update $I^{t+1}$ and double the number of membership constraints. Let $(C_1^*, \ldots, C_k^*)$ be the currently optimal partition. For each $l \in \{1, \ldots, k\}$ and $i \in C_l^*$ we compute

$$b_{li}^* = \frac{\mathrm{cut}\left(C_l^* \setminus \{i\}, \overline{C_l^*} \cup \{i\}\right)}{\hat{S}(C_l^* \setminus \{i\})} + \min_{s \neq l} \frac{\mathrm{cut}\left(C_s^* \cup \{i\}, \overline{C_s^*} \setminus \{i\}\right)}{\hat{S}(C_s^* \cup \{i\})} \tag{9}$$

and define $\mathcal{O}_l = \{(\pi_1, \ldots, \pi_{|C_l^*|}) \mid b_{l\pi_1}^* \geq b_{l\pi_2}^* \geq \ldots \geq b_{l\pi_{|C_l^*|}}^*\}$. The top-ranked vertices in $\mathcal{O}_l$ correspond to the ones which lead to the largest minimal increase in $\mathrm{BCut}$ when moved from $C_l^*$ to another component and thus are most likely to belong to their current component. Thus it is

natural to fix the top-ranked vertices for each component first. Note that the rankings $\mathcal{O}_l$, $l = 1, \ldots, k$ are updated when a better partition is found. Thus the membership constraints correspond always to the vertices which lead to largest minimal increase in $\mathrm{BCut}$ when moved to another component. In Figure 1 one can observe that the fixed labeled points are lying close to the centers of the found clusters. The number of membership constraints depends on the graph. The better separated the clusters are, the less membership constraints need to be enforced in order to avoid degenerate solutions. Finally, we stop the algorithm if we see no more improvement in the cut or the continuous objective and the continuous solution corresponds to a partition.

## 5 Experiments

We evaluate our method against a diverse selection of state-of-the-art clustering methods like spectral clustering (Spec) [7], BSpec [11], Graclus[1] [6], NMF based approaches PNMF [18], NSC [19], ONMF [20], LSD [21], NMFR [22] and MTV [13] which optimizes (5). We used the publicly available code [22, 13] with default settings. We run our method using 5 random initializations, 7 initializations based on the spectral clustering solution similar to [13] (who use 30 such initializations). In addition to the datasets provided in [13], we also selected a variety of datasets from the UCI repository shown below. For all the datasets not in [13], symmetric $k$-NN graphs are built with Gaussian weights $\exp\left(-\frac{s\|x-y\|^2}{\min\{\sigma_{x,k}^2, \sigma_{y,k}^2\}}\right)$, where $\sigma_{x,k}$ is the $k$-NN distance of point $x$. We chose the parameters $s$ and $k$ in a *method independent way* by testing for each dataset several graphs using all the methods over different choices of $k \in \{3, 5, 7, 10, 15, 20, 40, 60, 80, 100\}$ and $s \in \{0.1, 1, 4\}$. The best choice in terms of the clustering error across all the methods and datasets, is $s = 1, k = 15$.

|  | Iris | wine | vertebral | ecoli | 4moons | webkb4 | optdigits | USPS | pendigits | 20news | MNIST |
|---|---|---|---|---|---|---|---|---|---|---|---|
| # vertices | 150 | 178 | 310 | 336 | 4000 | 4196 | 5620 | 9298 | 10992 | 19928 | 70000 |
| # classes | 3 | 3 | 3 | 6 | 4 | 4 | 10 | 10 | 10 | 20 | 10 |

**Quantitative results:** In our first experiment we evaluate our method in terms of solving the balanced $k$-cut problem for various balancing functions, data sets and graph parameters. The following table reports the fraction of times a method achieves the best as well as strictly best balanced $k$-cut over all constructed graphs and datasets (in total 30 graphs per dataset). For reference, we also report the obtained cuts for other clustering methods although they do not directly minimize this criterion in *italic*; methods that directly optimize the criterion are shown in normal font. Our algorithm can handle all balancing functions and significantly outperforms all other methods across all criteria. For ratio and normalized cut cases we achieve better results than [7, 11, 6] which directly optimize this criterion. This shows that the greedy recursive bi-partitioning affects badly the performance of [11], which, otherwise, was shown to obtain the best cuts on several benchmark datasets [23]. This further shows the need for methods that directly minimize the multi-cut. It is striking that the competing method of [13], which directly minimizes the asymmetric ratio cut, is beaten significantly by Graclus as well as our method. As this clear trend is less visible in the qualitative experiments, we suspect that extreme graph parameters lead to fast convergence to a degenerate solution.

|  |  | Ours | MTV | BSpec | Spec | Graclus | PNMF | NSC | ONMF | LSD | NMFR |
|---|---|---|---|---|---|---|---|---|---|---|---|
| RCC-asym | Best (%) | **80.54** | 25.50 | *23.49* | *7.38* | *38.26* | *2.01* | *5.37* | *2.01* | *4.03* | *1.34* |
|  | Strictly Best (%) | **44.97** | 10.74 | *1.34* | *0.00* | *4.70* | *0.00* | *0.00* | *0.00* | *0.00* | *0.00* |
| RCC-sym | Best (%) | **94.63** | *8.72* | *19.46* | *6.71* | *37.58* | *0.67* | *4.03* | *0.00* | *0.67* | *0.67* |
|  | Strictly Best (%) | **61.74** | *0.00* | *0.67* | *0.00* | *4.70* | *0.00* | *0.00* | *0.00* | *0.00* | *0.00* |
| NCC-asym | Best (%) | **93.29** | *13.42* | *20.13* | *10.07* | *38.26* | *0.67* | *5.37* | *2.01* | *4.70* | *2.01* |
|  | Strictly Best (%) | **56.38** | *2.01* | *0.00* | *0.00* | *2.01* | *0.00* | *0.00* | *0.67* | *0.00* | *1.34* |
| NCC-sym | Best (%) | **98.66** | *10.07* | *20.81* | *9.40* | *40.27* | *1.34* | *4.03* | *0.67* | *3.36* | *1.34* |
|  | Strictly Best (%) | **59.06** | *0.00* | *0.00* | *0.00* | *1.34* | *0.00* | *0.00* | *0.00* | *0.00* | *0.00* |
| Rcut | Best (%) | **85.91** | *7.38* | *20.13* | *10.07* | *32.89* | *0.67* | *4.03* | *0.00* | *1.34* | *1.34* |
|  | Strictly Best (%) | **58.39** | *0.00* | *2.68* | *2.01* | *8.72* | *0.00* | *0.00* | *0.00* | *0.00* | *0.67* |
| Ncut | Best (%) | **95.97** | *10.07* | *20.13* | *9.40* | *37.58* | *1.34* | *4.70* | *0.67* | *3.36* | *0.67* |
|  | Strictly Best (%) | **61.07** | *0.00* | *0.00* | *0.00* | *4.03* | *0.00* | *0.00* | *0.00* | *0.00* | *0.00* |

**Qualitative results:** In the following table, we report the clustering errors and the balanced $k$-cuts obtained by all methods using the graphs built with $k = 15$, $s = 1$ for all datasets. As the main goal

is to compare to [13] we choose their balancing function (RCC-asym). Again, our method always achieved the best cuts across all datasets. In three cases, the best cut also corresponds to the best clustering performance. In case of vertebral, 20news, and webkb4 the best cuts actually result in high errors. However, we see in our next experiment that integrating ground-truth label information helps in these cases to improve the clustering performance significantly.

|  |  | Iris | wine | vertebral | ecoli | 4moons | webkb4 | optdigits | USPS | pendigits | 20news | MNIST |
|---|---|---|---|---|---|---|---|---|---|---|---|---|
| BSpec | Err(%) | 23.33 | 37.64 | 50.00 | 19.35 | 36.33 | 60.46 | 11.30 | 20.09 | 17.59 | 84.21 | 11.82 |
|  | BCut | **1.495** | 6.417 | **1.890** | 2.550 | 0.634 | **1.056** | 0.386 | 0.822 | 0.081 | 0.966 | 0.471 |
| Spec | Err(%) | **22.00** | 20.22 | 48.71 | **14.88** | 31.45 | 60.32 | 7.81 | 21.05 | 16.75 | 79.10 | 22.83 |
|  | BCut | 1.783 | 5.820 | 1.950 | 2.759 | 0.917 | 1.520 | 0.442 | 0.873 | 0.141 | 1.170 | 0.707 |
| PNMF | Err(%) | 22.67 | 27.53 | 50.00 | 16.37 | 35.23 | 60.94 | 10.37 | 24.07 | 17.93 | 66.00 | 12.80 |
|  | BCut | 1.508 | 4.916 | 2.250 | 2.652 | 0.737 | 3.520 | 0.548 | 1.180 | 0.415 | 2.924 | 0.934 |
| NSC | Err(%) | 23.33 | 17.98 | 50.00 | **14.88** | 32.05 | 59.49 | 8.24 | 20.53 | 19.81 | 78.86 | 21.27 |
|  | BCut | 1.518 | 5.140 | 2.046 | 2.754 | 0.933 | 3.566 | 0.482 | 0.850 | 0.101 | 2.233 | 0.688 |
| ONMF | Err(%) | 23.33 | 28.09 | 50.65 | 16.07 | 35.35 | 60.94 | 10.37 | 24.14 | 22.82 | 69.02 | 27.27 |
|  | BCut | 1.518 | 4.881 | 2.371 | 2.633 | 0.725 | 3.621 | 0.548 | 1.183 | 0.548 | 3.058 | 1.575 |
| LSD | Err(%) | 23.33 | 17.98 | 39.03 | 18.45 | 35.68 | 47.93 | 8.42 | 22.68 | 13.90 | 67.81 | 24.49 |
|  | BCut | 1.518 | 5.399 | 2.557 | 2.523 | 0.782 | 2.082 | 0.483 | 0.918 | 0.188 | 2.056 | 0.959 |
| NMFR | Err(%) | **22.00** | 11.24 | 38.06 | 22.92 | 36.33 | 40.73 | 2.08 | 22.17 | 13.13 | **39.97** | fail |
|  | BCut | 1.627 | 4.318 | 2.713 | 2.556 | 0.840 | 1.467 | 0.369 | 0.992 | 0.240 | 1.241 | - |
| Graclus | Err(%) | 23.33 | 8.43 | 49.68 | 16.37 | **0.45** | **39.97** | **1.67** | 19.75 | **10.93** | 60.69 | 2.43 |
|  | BCut | 1.534 | 4.293 | **1.890** | 2.414 | 0.589 | 1.581 | **0.350** | 0.815 | 0.092 | 1.431 | 0.440 |
| MTV | Err(%) | 22.67 | 18.54 | **34.52** | 22.02 | 7.72 | 48.40 | 4.11 | **15.13** | 20.55 | 72.18 | 3.77 |
|  | BCut | 1.508 | 5.556 | 2.433 | 2.500 | 0.774 | 2.346 | 0.374 | 0.940 | 0.193 | 3.291 | 0.458 |
| Ours | Err(%) | 23.33 | **6.74** | 50.00 | 16.96 | **0.45** | 60.46 | 1.71 | 19.72 | 19.95 | 79.51 | **2.37** |
|  | BCut | **1.495** | **4.168** | **1.890** | **2.399** | **0.589** | **1.056** | **0.350** | **0.802** | **0.079** | **0.895** | **0.439** |

**Transductive Setting:** We evaluate our method against [13] in a transductive setting. As in [13], we randomly sample either one label or a fixed percentage of labels per class from the ground truth. We report clustering errors and the cuts (RCC-asym) for both methods for different choices of labels. For label experiments their initialization strategy seems to work better as the cuts improve compared to the unlabeled case. However, observe that in some cases their method seems to fail completely (Iris and 4moons for one label per class).

| Labels |  |  | Iris | wine | vertebral | ecoli | 4moons | webkb4 | optdigits | USPS | pendigits | 20news | MNIST |
|---|---|---|---|---|---|---|---|---|---|---|---|---|---|---|
| 1 | MTV | Err(%) | 33.33 | 9.55 | **42.26** | 13.99 | 35.75 | 51.98 | **1.69** | **12.91** | 14.49 | **50.96** | 2.45 |
|  |  | BCut | 3.855 | 4.288 | **2.244** | **2.430** | 0.723 | 1.596 | **0.352** | 0.846 | 0.127 | 1.286 | **0.439** |
|  | Ours | Err(%) | **22.67** | **8.99** | 50.32 | 15.48 | **0.57** | **45.11** | **1.69** | 12.98 | **10.98** | 68.53 | **2.36** |
|  |  | BCut | **1.571** | **4.234** | 2.265 | 2.432 | **0.610** | **1.471** | **0.352** | **0.812** | **0.113** | **1.057** | 0.439 |
| 1% | MTV | Err(%) | 33.33 | 10.67 | **39.03** | 14.29 | **0.45** | 48.38 | **1.67** | 5.21 | **7.75** | 40.18 | 2.41 |
|  |  | BCut | 3.855 | 4.277 | 2.300 | 2.429 | **0.589** | 1.584 | **0.354** | 0.789 | 0.129 | 1.208 | 0.443 |
|  | Ours | Err(%) | **22.67** | **6.18** | 41.29 | **13.99** | **0.45** | **41.63** | **1.67** | **5.13** | **7.75** | **37.42** | **2.33** |
|  |  | BCut | **1.571** | **4.220** | 2.288 | 2.419 | **0.589** | **1.462** | **0.354** | 0.789 | **0.128** | **1.157** | **0.442** |
| 5% | MTV | Err(%) | **17.33** | 7.87 | 40.65 | 14.58 | **0.45** | 40.09 | **1.51** | **4.85** | 1.79 | 31.89 | **2.18** |
|  |  | BCut | **1.685** | 4.330 | **2.701** | 2.462 | **0.589** | 1.763 | 0.369 | 0.812 | 0.188 | 1.254 | 0.455 |
|  | Ours | Err(%) | **17.33** | **6.74** | **37.10** | **13.99** | **0.45** | **38.04** | 1.53 | **4.85** | **1.76** | **30.07** | 2.18 |
|  |  | BCut | **1.685** | **4.224** | 2.724 | **2.461** | **0.589** | **1.719** | 0.369 | **0.811** | 0.188 | **1.210** | 0.455 |
| 10% | MTV | Err(%) | 18.67 | 7.30 | 39.03 | 13.39 | **0.38** | **40.63** | **1.41** | **4.19** | **1.24** | 27.80 | 2.03 |
|  |  | BCut | **1.954** | 4.332 | 3.187 | **2.776** | 0.592 | 2.057 | **0.377** | 0.833 | **0.197** | 1.346 | 0.465 |
|  | Ours | Err(%) | **14.67** | **6.74** | **33.87** | **13.10** | **0.38** | 41.97 | **1.41** | 4.25 | **1.24** | **26.55** | **2.02** |
|  |  | BCut | 1.960 | **4.194** | **3.134** | 2.778 | 0.592 | **1.972** | **0.377** | 0.833 | **0.197** | **1.314** | 0.465 |

# 6 Conclusion

We presented a framework for directly minimizing the balanced $k$-cut problem based on a new tight continuous relaxation. Apart from the standard ratio/normalized cut, our method can also handle new application-specific balancing functions. Moreover, in contrast to a recursive splitting approach [24], our method enables the direct integration of prior information available in form of must/cannot-link constraints, which is an interesting topic for future research. Finally, the monotonic descent algorithm proposed for the difficult sum-of-ratios problem is another key contribution of the paper that is of independent interest.

## Footnotes

[1] Since [6], a multi-level algorithm directly minimizing Rcut/Ncut, is shown to be superior to METIS [17], we do not compare with [17].

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
