[Supplementary Material]

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

}: \quad F_{(i)} \in \Delta_k, \qquad i = 1, \ldots, n, \quad \text{(simplex constraints)}
$$
$$
\max\{F_{(i)}\} = 1, \quad \forall i \in I, \qquad \text{(membership constraints)}
$$
$$
S(F_l) \geq m, \qquad l = 1, \ldots, k, \quad \text{(size constraints)}
$$

where $S$ is the Lovasz extension of the set function $\hat{S}$ and $m = \min_{C \subsetneq V, \, C \neq \emptyset} \hat{S}(C)$. We have $m = 1$, for *Ratio Cut* and *Ratio Cheeger Cut* whereas $m = k - 1$ for *Asymmetric Ratio Cheeger Cut*. Note that TV is the Lovasz extension of the cut functional $\mathrm{cut}(C, \overline{C})$. In order to simplify notation we denote for a matrix $F \in \mathbb{R}^{n \times k}$ by $F_l$ the $l$-th column of $F$ and by $F_{(i)}$ the $i$-th row of $F$. Note that the rows of $F$ correspond to the vertices of the graph and the $j$-th column of $F$ corresponds to the set $C_j$ of the desired partition. The set $I \subset V$ in the membership constraints is chosen adaptively by our method during the sequential optimization described in Section 4.

An obvious question is how to get from the continuous solution $F^*$ of (3) to a partition $(C_1, \ldots, C_k) \in P_k$ which is typically called *rounding*. Given $F^*$ we construct the sets, by assigning each vertex $i$ to the column where the $i$-th row attains its maximum. Formally,

$$
C_i = \{j \in V \mid i = \arg\max_{s=1,\ldots,k} F_{js}\}, \quad i = 1, \ldots, k, \quad \text{(Rounding)} \tag{4}
$$

where ties are broken randomly. If there exists a row such that the rounding is not unique, we say that the solution is weakly degenerated. If furthermore the resulting set $(C_1, \ldots, C_k)$ do not form a partition, that is one of the sets is empty, then we say that the solution is strongly degenerated.

First, we connect our relaxation to the previous work of [11] for the case $k = 2$. Indeed for symmetric balancing function such as the *Ratio Cheeger Cut*, our continuous relaxation (3) is exact even without membership and size constraints.

**Theorem 1** *Let $\hat{S}$ be a non-negative symmetric balancing function, $\hat{S}(C) = \hat{S}(\overline{C})$, and denote by $p^*$ the optimal value of* (3) *without membership and size constraints for $k = 2$. Then it holds*

$$
p^* = \min_{(C_1, C_2) \in P_2} \sum_{i=1}^{2} \frac{\mathrm{cut}(C_i, \overline{C_i})}{\hat{S}(C_i)}.
$$

*Furthermore there exists a solution $F^*$ of* (3) *such that $F^* = [\mathbf{1}_{C^*}, \mathbf{1}_{\overline{C^*}}]$, where $(C^*, \overline{C^*})$ is the optimal balanced 2-cut partition.*

**Proof:** Note that $\mathrm{cut}(C, \overline{C})$ is a symmetric set function and $\hat{S}$ by assumption. Thus with $C_2 = \overline{C_1}$,

$$
\frac{\mathrm{cut}(C_1, \overline{C_1})}{\hat{S}(C_1)} + \frac{\mathrm{cut}(C_2, \overline{C_2})}{\hat{S}(C_2)} = 2 \frac{\mathrm{cut}(C_1, \overline{C_1})}{\hat{S}(C_1)}.
$$

Moreover, as $\mathrm{TV}(\alpha f + \beta \mathbf{1}) = |\alpha| \, \mathrm{TV}(f)$ and by symmetry of $\hat{S}$ also $S(\alpha f + \beta \mathbf{1}) = |\alpha| \, S(f)$ (see [14, 11]). The simplex constraint implies that $F_2 = \mathbf{1} - F_1$ and thus

$$
\frac{\mathrm{TV}(F_2)}{S(F_2)} = \frac{\mathrm{TV}(\mathbf{1} - F_1)}{S(\mathbf{1} - F_1)} = \frac{\mathrm{TV}(F_1)}{S(F_1)}.
$$

Thus we can write problem (3) equivalently as

$$\min_{f \in [0,1]^V} 2\frac{\mathrm{TV}(f)}{S(f)}.$$

As for all $A \subset V$, $\mathrm{TV}(\mathbf{1}_A) = \mathrm{cut}(A, \overline{A})$ and $S(\mathbf{1}_A) = \hat{S}(A)$, we have

$$\min_{f \in [0,1]^V} \frac{\mathrm{TV}(f)}{S(f)} \leq \min_{C \subset V} \frac{\mathrm{cut}(C, \overline{C})}{\hat{S}(C)}.$$

However, it has been shown in [11] that $\min_{f \in \mathbb{R}^V} \frac{\mathrm{TV}(f)}{S(f)} = \min_{C \subset V} \frac{\mathrm{cut}(C, \overline{C})}{\hat{S}(C)}$ and that there exists a continuous solution such that $f^* = \mathbf{1}_{C^*}$, where $C^* = \arg\min_{C \subset V} \frac{\mathrm{cut}(C, \overline{C})}{\hat{S}(C)}$. As $F^* = [f^*, \mathbf{1} - f^*] = [\mathbf{1}_{C^*}, \mathbf{1}_{\overline{C^*}}]$ this finishes the proof. $\qquad\square$

Note that rounding trivially yields a solution in the setting of the previous theorem.

A second result shows that indeed our proposed optimization problem (3) is a relaxation of the balanced $k$-cut problem (1). Furthermore, the relaxation is exact if $I = V$.

**Proposition 1** *The continuous problem* (3) *is a relaxation of the $k$-cut problem* (1). *The relaxation is exact, i.e., both problems are equivalent, if $I = V$.*

**Proof:** For any $k$-way partition $(C_1, \ldots, C_k)$, we can construct $F = (\mathbf{1}_{C_1}, \ldots, \mathbf{1}_{C_k})$. It obviously satisfies the membership and size constraints and the simplex constraint is satisfied as $\cup_i C_i = V$ and $C_i \cap C_j = \emptyset$ if $i \neq j$. Thus $F$ is feasible for problem (3) and has the same objective value because

$$\mathrm{TV}(\mathbf{1}_C) = \mathrm{cut}(C, \overline{C}), \quad S(\mathbf{1}_C) = \hat{S}(C).$$

Thus problem (3) is a relaxation of (1).

If $I = V$, then the simplex together with the membership constraints imply that each row $F_{(i)}$ contains exactly one non-zero element which equals 1, i.e., $F \in \{0, 1\}^{n \times k}$. Define for $l = 1, \ldots, k$, $C_l = \{i \in V \mid F_{il} = 1\}$ (i.e, $F_l = \mathbf{1}_{C_l}$), then it holds $\cup_l C_l = V$ and $C_l \cap C_j = \emptyset$, $l \neq j$. From the size constraints, we have for $l = 1, \ldots, k$, $0 < m \leq S(F_l) = S(\mathbf{1}_{C_l}) = \hat{S}(C_l)$. Thus $\hat{S}(C_l) > 0$, $l = 1, \ldots, k$, which by assumption on $\hat{S}$ implies that each $C_l$ is non-empty. Hence the only feasible points allowed are indicators of $k$-way partitions and the equivalence of (1) and (3) follows. $\qquad\square$

The row-wise simplex and membership constraints enforce that each vertex in $I$ belongs to exactly one component. Note that these constraints alone (even if $I = V$) can still not guarantee that $F$ corresponds to a $k$-way partition since an entire column of $F$ can be zero. This is avoided by the column-wise size constraints that enforce that each component has at least one vertex.

If $I = V$ it is immediate from the proof that problem (3) is no longer a continuous problem as the feasible set are only indicator matrices of partitions. In this case rounding yields trivially a partition. On the other hand, if $I = \emptyset$ (i.e., no membership constraints), and $k > 2$ it is not guaranteed that rounding of the solution of the continuous problem yields a partition. Indeed, we will see in the following that for symmetric balancing functions one can, under these conditions, show that the solution is always strongly degenerated and rounding does not yield a partition (see Theorem 2). Thus we observe that the index set $I$ controls the degree to which the partition constraint is enforced. The idea behind our suggested relaxation is that it is well known in image processing that minimizing the total variation yields piecewise constant solutions (in fact this follows from seeing the total variation as Lovasz extension of the cut). Thus if $|I|$ is sufficiently large, the vertices where the values are fixed to 0 or 1 propagate this to their neighboring vertices and finally to the whole graph. We discuss the choice of $I$ in more detail in Section 4.

**Simplex constraints alone are not sufficient to yield a partition:** Our approach has been inspired by [13] who proposed the following continuous relaxation for the *Asymmetric Ratio Cheeger Cut*

$$\min_{\substack{F=(F_1,\dots,F_k),\\ F\in\mathbb{R}_+^{n\times k}}} \sum_{l=1}^{k} \frac{\mathrm{TV}(F_l)}{\left\| F_l - \mathrm{quant}_{k-1}(F_l)\right\|_1} \tag{5}$$

$$\text{subject to}: F_{(i)} \in \Delta_k, \quad i=1,\dots,n, \quad \text{(simplex constraints)}$$

where $S(f) = \left\| f - \mathrm{quant}_{k-1}(f)\right\|_1$ is the Lovasz extension of $\hat{S}(C) = \min\{(k-1)|C|,\overline{C}\}$ and $\mathrm{quant}_{k-1}(f)$ is the $k-1$-quantile of $f\in\mathbb{R}^n$. Note that in their approach no membership constraints and size constraints are present.

We now show that the usage of simplex constraints in the optimization problem (3) is not sufficient to guarantee that the solution $F^*$ can be rounded to a partition for any symmetric balancing function in (1). For asymmetric balancing functions as employed for the *Asymmetric Ratio Cheeger Cut* by [13] in their relaxation (5) we can prove such a strong result only in the case where the graph is disconnected. However, note that if the number of components of the graph is less than the number of desired clusters $k$, the multi-cut problem is still non-trivial.

**Theorem 2** *Let $\hat{S}(C)$ be any non-negative symmetric balancing function. Then the continuous relaxation*

$$\min_{\substack{F=(F_1,\dots,F_k),\\ F\in\mathbb{R}_+^{n\times k}}} \sum_{l=1}^{k} \frac{\mathrm{TV}(F_l)}{S(F_l)} \tag{6}$$

$$\text{subject to}: F_{(i)} \in \Delta_k, \quad i=1,\dots,n, \quad \text{(simplex constraints)}$$

*of the balanced $k$-cut problem* (1) *is void in the sense that the optimal solution $F^*$ of the continuous problem can be constructed from the optimal solution of the 2-cut problem and $F^*$ cannot be rounded into a $k$-way partition, see* (4). *If the graph is disconnected, then the same holds also for any non-negative asymmetric balancing function.*

**Proof:** First, we derive a lower bound on the optimum of the continuous relaxation (6). Then we construct a feasible point for (6) that achieves this lower bound but cannot yield a partitioning thus finishing the proof.

Let $(C^*, \overline{C^*}) = \arg\min_{C\subset V} \frac{\mathrm{cut}(C,\overline{C})}{\hat{S}(C)}$ be an optimal 2-way partition for the given graph. Using the exact relaxation result for the balanced 2-cut problem in Theorem 3.1. in [11], we have

$$\min_{F:F_{(i)}\in\Delta_k} \sum_{l=1}^{k} \frac{\mathrm{TV}(F_l)}{S(F_l)} \geq \sum_{l=1}^{k} \min_{f\in\mathbb{R}^n} \frac{\mathrm{TV}(f)}{S(f)} = \sum_{l=1}^{k} \min_{C\subset V} \frac{\mathrm{cut}(C,\overline{C})}{\hat{S}(C)} = k\,\frac{\mathrm{cut}(C^*,\overline{C^*})}{\hat{S}(C^*)}.$$

Now define $F_1 = \mathbf{1}_{C^*}$ and $F_l = \alpha_l \mathbf{1}_{\overline{C^*}}$, $l = 2,\dots,k$ such that $\sum_{l=2}^{k}\alpha_l = 1, \alpha_l > 0$. Clearly $F = (F_1,\dots,F_k)$ is feasible for the problem (6) and the corresponding objective value is

$$\frac{\mathrm{TV}(\mathbf{1}_{C^*})}{S(\mathbf{1}_{C^*})} + \sum_{l=2}^{k} \frac{\alpha_l \mathrm{TV}(\mathbf{1}_{\overline{C^*}})}{\alpha_l S(\mathbf{1}_{\overline{C^*}})} = \sum_{l=1}^{k} \frac{\mathrm{cut}(C^*,\overline{C^*})}{\hat{S}(C^*)},$$

where we used the 1-homogeneity of TV and $S$ [14] and the symmetry of $\mathrm{cut}$ and $\hat{S}$.

Thus the solution $F$ constructed as above from the 2-cut problem is indeed optimal for the continuous relaxation (6) and it is not possible to obtain a $k$-way partition from this solution as there will be $k-2$ sets that are empty. Finally, the argument can be extended to asymmetric set functions if there exists a set $C$ such that $\mathrm{cut}(C,\overline{C}) = 0$ as in this case it does not matter that $\hat{S}(C) \neq \hat{S}(\overline{C})$ in order that the argument holds. $\square$

The proof of Theorem 2 shows additionally that for any balancing function if the graph is disconnected, the solution of the continuous relaxation (6) is always zero, while clearly the solution of the balanced $k$-cut problem need not be zero. This shows that the relaxation can be arbitrarily bad in this case. In fact the relaxation for the asymmetric case can even fail if the graph is not disconnected but there exists a cut of the graph which is very small as the following corollary indicates.

Figure 1: Toy example illustrating that the relaxation of [13] converges to a degenerate solution when applied to a graph with dominating 2-cut. (a) 10NN-graph generated from three Gaussians in 10 dimensions (b) continuous solution of (5) from [13] for $k = 3$, (c) rounding of the continuous solution of [13] does not yield a 3-partition (d) continuous solution found by our method together with the vertices $i \in I$ (black) where the membership constraint is enforced. Our continuous solution corresponds already to a partition. (e) clustering found by rounding of our continuous solution (trivial as we have converged to a partition). In (b)-(e), we color data point $i$ according to $F_{(i)} \in \mathbb{R}^3$.

**Corollary 1** *Let $\hat{S}$ be an asymmetric balancing function and $C^* = \arg\min\limits_{C \subset V} \frac{\text{cut}(C, \overline{C})}{\hat{S}(C)}$ and suppose that $\phi^* := (k-1)\frac{\text{cut}(C^*, \overline{C^*})}{\hat{S}(C^*)} + \frac{\text{cut}(C^*, \overline{C^*})}{\hat{S}(\overline{C^*})} < \min_{(C_1,\ldots,C_k) \in P_k} \sum_{i=1}^{k} \frac{\text{cut}(C_i, \overline{C_i})}{\hat{S}(C_i)}$. Then there exists a feasible $F$ with $F_1 = \mathbf{1}_{\overline{C^*}}$ and $F_l = \alpha_l \mathbf{1}_{C^*}$, $l = 2,\ldots,k$ such that $\sum_{l=2}^{k} \alpha_l = 1, \alpha_l > 0$ for (6) which has objective $\sum_{i=1}^{k} \frac{\text{TV}(F_i)}{S(F_i)} = \phi^*$ and which cannot be rounded to a k-way partition.*

**Proof:** Let $F_1 = \mathbf{1}_{\overline{C^*}}$ and $F_l = \alpha_l \mathbf{1}_{C^*}$, $l = 2,\ldots,k$ such that $\sum_{l=2}^{k} \alpha_l = 1, \alpha_l > 0$. Clearly $F = (F_1,\ldots,F_k)$ is feasible for the problem (6) and the corresponding objective value is

$$\sum_{l=1}^{k} \frac{\text{TV}(F_l)}{S(F_l)} = \frac{\text{TV}(\mathbf{1}_{\overline{C^*}})}{S(\mathbf{1}_{\overline{C^*}})} + \sum_{l=2}^{k} \frac{\alpha_l \text{TV}(\mathbf{1}_{C^*})}{\alpha_l S(\mathbf{1}_{C^*})}$$
$$= \frac{\text{cut}(C^*, \overline{C^*})}{\hat{S}(\overline{C^*})} + (k-1)\frac{\text{cut}(C^*, \overline{C^*})}{\hat{S}(C^*)},$$

where we used the 1-homogeneity of TV and $S$ [14] and the symmetry of cut. This $

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

**Proof:** Let $(F^{t+1}, \delta^{+,\,t+1}, \delta^{-,\,t+1})$ be the optimal solution of the inner problem (8). By the feasibility of $(F^{t+1}, \delta^{+,\,t+1}, \delta^{-,\,t+1})$ and $S(F_l^{t+1}) \geq m$,

$$\frac{\mathrm{TV}(F_l^{t+1})}{S(F_l^{t+1})} \leq \frac{\lambda_l^t \left\langle s_l^t, F_l^{t+1} \right\rangle + m\delta_l^{+,\,t+1} - M\delta_l^{-,\,t+1}}{S(F_l^{t+1})}$$

$$\leq \lambda_l^t + \frac{m\delta_l^{+,\,t+1} - M\delta_l^{-,\,t+1}}{S(F_l^{t+1})} \leq \lambda_l^t + \delta_l^{+,\,t+1} - \delta_l^{-,\,t+1}$$

Summing over all ratios, we have

$$\sum_{l=1}^{k} \frac{\mathrm{TV}(F_l^{t+1})}{S(F_l^{t+1})} \leq \sum_{l=1}^{k} \lambda_l^t + \sum_{l=1}^{k} \delta_l^{+,\,t+1} - \delta_l^{-,\,t+1}$$

Noting that $\delta_l^+ = \delta_l^- = 0$, $F = F^t$ is feasible for (8), the optimal value $\sum_{l=1}^{k} \delta_l^{+,\,t+1} - \delta_l^{-,\,t+1}$ has to be either strictly negative in which case we have strict descent

$$\sum_{l=1}^{k} \frac{\mathrm{TV}(F_l^{t+1})}{S(F_l^{t+1})} < \sum_{l=1}^{k} \lambda_l^t$$

or the previous iterate $F^t$ together with $\delta_l^+ = \delta_l^- = 0$ is already optimal and hence the algorithm terminates. □

The inner problem (8) is convex, but contains the non-smooth term TV in the constraints. We eliminate the non-smoothness by introducing additional variables and derive an equivalent linear programming (LP) formulation. We solve this LP via the PDHG algorithm [15, 16]. The LP and the exact iterates can be found in the supplementary material.

**Lemma 1** *The convex inner problem* (8) *is equivalent to the following linear optimization problem where $E$ is the set of edges of the graph and $w \in \mathbb{R}^{|E|}$ are the edge weights.*

$$\min_{\substack{F \in \mathbb{R}_+^{n \times k}, \\ \alpha \in \mathbb{R}_+^{|E| \times k}, \\ \delta^+ \in \mathbb{R}_+^k,\ \delta^- \in \mathbb{R}_+^k}} \quad \sum_{l=1}^{k} \delta_l^+ - \delta_l^- \tag{9}$$

$$\begin{aligned}
\text{subject to}: \ & \langle w, \alpha_l \rangle \leq \lambda_l^t \left\langle s_l^t, F_l \right\rangle + \delta_l^+ m - \delta_l^- M, && l = 1, \ldots, k, && \textit{(descent constraints)}\\
& F_{(i)} \in \Delta_k, && i = 1, \ldots, n, && \textit{(simplex constraints)}\\
& F_{ij_i} = 1, && \forall (i, j_i) \in L, && \textit{(label constraints)}\\
& \left\langle s_l^t, F_l^t \right\rangle \geq m, && l = 1, \ldots, k, && \textit{(size constraints)}\\
& -(\alpha_l)_{ij} \leq F_{il} - F_{jl} \leq (\alpha_l)_{ij}, && l = 1, \ldots, k, && \forall (i, j) \in E.
\end{aligned}$$

**Proof:** We define new variables $\alpha_l \in \mathbb{R}^{|E|}$ for each column $l$ and introduce constraints $(\alpha_l)_{ij} = |(F_l)_i - (F_l)_j)|$, which allows us to rewrite $\mathrm{TV}(F_l)$ as $\langle w, \alpha_l \rangle$. These equality constraints can be replaced by the inequality constraints $(\alpha_l)_{ij} \geq |(f_l)_i - (f_l)_j)|$ without changing the optimality of the problem, because at the optimal these constraints are active. Otherwise one can decrease $(\alpha_l)_{ij}$ while still being feasible since $w$ is non-negative. Finally, these inequality constraints are rewritten using the fact that $|x| \leq y \Leftrightarrow -y \leq x \leq y$, for $y \geq 0$. □

### 4.0.1  Solving LP via PDHG

Recently, first-order primal-dual hybrid gradient descent (PDHG for short) methods have been proposed [17, 15] to efficiently solve a class of convex optimization problems that can be rewritten as

the following saddle-point problem

$$\min_{x \in X} \max_{y \in Y} \langle Ax, y \rangle + G(x) - \Phi^*(y),$$

where $X$ and $Y$ are finite-dimensional vector spaces and $A : X \to Y$ is a linear operator and $G$ and $\Phi^*$ are convex functions. It has been shown that the PDHG algorithm achieves good performance in solving huge linear programming problems that appear in computer vision applications. We now show how the linear programming problem

$$\min_{x \geq 0} \; \langle c, x \rangle$$
$$\text{subject to} : A_1 x \leq b_1$$
$$A_2 x = b_2$$

can be rewritten as a saddle-point problem so that PDHG can be applied.

By introducing the Lagrange multipliers $y$, the optimal value of the LP can be written as

$$\min_{x \geq 0} \langle c, x \rangle + \max_{y_1 \geq 0, \, y_2} \langle y_1, A_1 x - b_1 \rangle + \langle y_2, A_2 x - b_2 \rangle$$

$$= \min_{x} \max_{y_1, \, y_2} \langle c, x \rangle + \iota_{x \geq 0}(x) + \langle y_1, A_1 x \rangle + \langle y_2, A_2 x \rangle - \langle b_1, y_1 \rangle - \langle b_2, y_2 \rangle + \iota_{y_1 \geq 0}(y_1),$$

where $\iota_{\cdot \geq 0}$ is the indicator function that takes a value of 0 on the non-negative orthant and $\infty$ elsewhere.

Define $b = \begin{pmatrix} b_1 \\ b_2 \end{pmatrix}$, $A = \begin{pmatrix} A_1 \\ A_2 \end{pmatrix}$ and $y = \begin{pmatrix} y_1 \\ y_2 \end{pmatrix}$. Then the saddle point problem corresponding to the LP is given by

$$\min_{x} \max_{y_1, \, y_2} \langle c, x \rangle + + \iota_{x \geq 0}(x) + \langle y, Ax \rangle - \langle b, y \rangle + \iota_{y_1 \geq 0}(y_1).$$

The primal and dual iterates for this saddle-point problem can be obtained as

$$x^{r+1} = \max\{0, x^r - \tau(A^T y^r + c)\},$$
$$y_1^{r+1} = \max\{0, y_1^r + \sigma(A_1 \bar{x}^{r+1} - b_1)\},$$
$$y_2^{r+1} = y_2^r + \sigma(A_2 \bar{x}^{r+1} - b_2),$$

where $\bar{x}^{r+1} = 2x^{r+1} - x^r$. Here the primal and dual step sizes $\tau$ and $\sigma$ are chosen such that $\tau \sigma \|A\|^2 < 1$, where $\|.\|$ denotes the operator norm.

Instead of the global step sizes $\tau$ and $\sigma$, we use in our implementation the diagonal preconditioning matrices introduced in [16] as it is shown to improve the practical performance of PDHG. The diagonal elements of these preconditioning matrices $\boldsymbol{\tau}$ and $\boldsymbol{\sigma}$ are given by

$$\boldsymbol{\tau}_j = \frac{1}{\sum_{i=1}^{n_r} |A_{ij}|}, \forall j \in \{1, \dots, n_c\}, \; \boldsymbol{\sigma}_i = \frac{1}{\sum_{i=1}^{n_c} |A_{ij}|}, \forall i \in \{1, \dots, n_r\},$$

where $n_r$, $n_c$ are the number of rows and the number of columns of the matrix $A$.

For completeness, we now present the explicit form of the primal and dual iterates of the preconditioned PDHG for the LP (9). Let $\theta \in \mathbb{R}^k$, $\mu \in \mathbb{R}^n$, $\zeta \in \mathbb{R}^{|L|}$, $\nu \in \mathbb{R}^k$, $\eta_l \in \mathbb{R}^{|E|}$, $\xi_l \in \mathbb{R}^{|E|}$, $\forall l \in \{1, \dots, k\}$ be the Lagrange multipliers corresponding to the descent, simplex, label, size and the two sets of additional constraints (introduced to eliminate the non-smoothness) respectively. Let $B : \mathbb{R}^{|E|} \to \mathbb{R}^{|V|}$ be a linear mapping defined as $(Bz)_i = \sum_{j:(i,j)\in E} z_{ij} - z_{ji}$ and $\mathbf{1}_n \in \mathbb{R}^n$ denote a vector of all ones. Then the primal iterates for the LP (9) are given by

$$F_l^{r+1} = \max \left\{ 0, F_l^r - \boldsymbol{\tau}_{F, \, l} \left( (-\theta_l^r \lambda_l^t - \nu_l^r) s_l^t + \mu^r + Z_l^r + B(\eta_l^r - \xi_l^r) \right) \right\}, \; \forall l \in \{1, \dots, k\},$$

$$\alpha_l^{r+1} = \max \left\{ 0, \alpha_l^r - \boldsymbol{\tau}_{\alpha, \, l} \left( \theta_l^r w - \eta_l^r - \xi_l^r \right) \right\}, \; \forall l \in \{1, \dots, k\},$$

$$\delta^{+, \, r+1} = \max \left\{ 0, \delta^{+, \, r} - \boldsymbol{\tau}_{\delta^+} \left( -m\theta^r + \mathbf{1}_k \right) \right\},$$

$$\delta^{-, \, r+1} = \max \left\{ 0, \delta^{-, \, r} - \boldsymbol{\tau}_{\delta^-} \left( M\theta^r - \mathbf{1}_k \right) \right\},$$

where $Z_l^r \in \mathbb{R}^n, l = 1, \ldots, k$, are given by $(Z_l^r)_i = \zeta_{il}^r$, if $(i,l) \in L$ and 0 otherwise. Here $\boldsymbol{\tau}_{F,\,l}$, $\boldsymbol{\tau}_{\alpha,\,l}$, $\boldsymbol{\tau}_{\delta^+}$, $\boldsymbol{\tau}_{\delta^-}$ are the diagonal preconditioning matrices whose diagonal elements are given by

$$(\boldsymbol{\tau}_{F,\,l})_i = \frac{1}{(1 + \lambda_l^t)\,|(s_l^t)_i| + 2d_i + \rho_{il} + 1},\ \forall i \in \{1, \ldots, n\},$$

$$(\boldsymbol{\tau}_{\alpha,\,l})_{ij} = \frac{1}{w_{ij} + 2},\ \forall (i,j) \in E,$$

$$(\boldsymbol{\tau}_{\delta^+})_l = \frac{1}{m},\ \forall l \in \{1, \ldots, k\},$$

$$(\boldsymbol{\tau}_{\delta^-})_l = \frac{1}{M},\ \forall l \in \{1, \ldots, k\},$$

where $d_i$ is the number of vertices adjacent to the $i^{th}$ vertex and $\rho_{il} = 1$, if $(i,l) \in L$ and 0 otherwise.

The dual iterates are given by

$$\theta_l^{r+1} = \max\left\{0, \theta_l^r + \boldsymbol{\sigma}_{\theta,\,l}\Big( \langle w, \bar{\alpha}_l^{r+1} \rangle - \lambda_l^t \langle s_l^t, \bar{F}_l^{r+1} \rangle - m\bar{\delta}_l^{+,\,r+1} + M\bar{\delta}_l^{-,\,r+1} \Big) \right\},\ l = 1, \ldots, k,$$

$$\mu^{r+1} = \mu^r + \boldsymbol{\sigma}_\mu\Big( \bar{F}^{r+1}\mathbf{1}_k - \mathbf{1}_n \Big),$$

$$\zeta_{il}^{r+1} = \zeta_{il}^r + \boldsymbol{\sigma}_\zeta\Big( \bar{F}_{il}^{r+1} - 1 \Big),\ \forall (i,l) \in L,$$

$$\nu_l^{r+1} = \max\left\{0, \nu_l^r + \boldsymbol{\sigma}_{\nu,\,l}\Big( -\langle s_l^t, \bar{F}_l^{r+1} \rangle + m \Big) \right\},\ \forall l \in \{1, \ldots, k\},$$

$$\eta_l^{r+1} = \max\left\{0, \eta_l^r + \boldsymbol{\sigma}_{\eta,\,l}\Big( -\bar{\alpha}_l^{r+1} + \bar{F}_{il}^{r+1} - \bar{F}_{jl}^{r+1} \Big) \right\},\ \forall l \in \{1, \ldots, k\},$$

$$\xi_l^{r+1} = \max\left\{0, \xi_l^r + \boldsymbol{\sigma}_{\xi,\,l}\Big( -\bar{\alpha}_l^{r+1} - \bar{F}_{il}^{r+1} + \bar{F}_{jl}^{r+1} \Big) \right\},\ \forall l \in \{1, \ldots, k\},$$

where

$$\boldsymbol{\sigma}_{\theta,\,l} = \frac{1}{\langle w, 1 \rangle + \lambda_l^t \sum_{i=1}^n |(s_l^t)_i| + m + M},\ \boldsymbol{\sigma}_\zeta = 1,\ \boldsymbol{\sigma}_{\nu,\,l} = \frac{1}{\sum_{i=1}^n |(s_l^t)_i|},$$

and $\boldsymbol{\sigma}_\mu$, $\boldsymbol{\sigma}_{\eta,1}$, $\boldsymbol{\sigma}_{\xi,l}$ are the diagonal preconditioning matrices whose diagonal elements are given by

$$(\boldsymbol{\sigma}_\mu)_i = \frac{1}{k},\ \forall i \in \{1, \ldots, n\},\ (\boldsymbol{\sigma}_{\eta,l})_{ij} = (\boldsymbol{\sigma}_{\xi,l})_{ij} = \frac{1}{3},\ \forall (i,j) \in E.$$

From the iterates, one sees that the computational cost per iteration is $O(|E|)$. In our implementation, we further reformulated the LP (9) by directly integrating the label constraints, thereby reducing the problem size and getting rid of the dual variable $\zeta$.

## 4.1 Choice of membership constraints $I$

The overall algorithm scheme for solving the problem (1) is given in the supplementary material. For the membership constraints we start initially with $I^0 = \emptyset$ and sequentially solve the inner problem (8). From its solution $F^{t+1}$ we construct a $P_k' = (C_1, \ldots, C_k)$ via rounding, see (4). We repeat this process until we either do not improve the resulting balanced $k$-cut or $P_k'$ is not a partition. In this case we update $I^{t+1}$ and double the number of membership constraints. Let $(C_1^*, \ldots, C_k^*)$ be the currently optimal partition. For each $l \in \{1, \ldots, k\}$ and $i \in C_l^*$ we compute

$$b_{li}^* = \frac{\mathrm{cut}\big(C_l^* \backslash \{i\}, \overline{C_l^*} \cup \{i\}\big)}{\hat{S}(C_l^* \backslash \{i\})} + \min_{s \neq l} \frac{\mathrm{cut}\big(C_s^* \cup \{i\}, \overline{C_s^*} \backslash \{i\}\big)}{\hat{S}(C_s^* \cup \{i\})} \tag{10}$$

and define $\mathcal{O}_l = \{(\pi_1, \ldots, \pi_{|C_l^*|}) \,|\, b_{l\pi_1}^* \geq b_{l\pi_2}^* \geq \ldots \geq b_{l\pi_{|C_l^*|}}^*\}$. The top-ranked vertices in $\mathcal{O}_l$ correspond to the ones which lead to the largest minimal increase in BCut when moved from $C_l^*$ to another component and thus are most likely to belong to their current component. Thus it is natural to fix the top-ranked vertices for each component first. Note that the rankings $\mathcal{O}_l, l =$

$1, \ldots, k$ are updated when a better partition is found. Thus the membership constraints correspond always to the vertices which lead to largest minimal increase in $\mathrm{BCut}$ when moved to another component. In Figure 1 one can observe that the fixed labeled points are lying close to the centers of the found clusters. The number of membership constraints depends on the graph. The better separated the clusters are, the less membership constraints need to be enforced in order to avoid degenerate solutions. Finally, we stop the algorithm if we see no more improvement in the cut or the continuous objective and the continuous solution corresponds to a partition.

---
**Algorithm 1 for solving** (1)

---
1: **Initialization:** $F^0 \in \mathbb{R}_+^{n \times k}$ be such that $F^0 \mathbf{1}_k = \mathbf{1}_n$, $\lambda_l^0 = \frac{\mathrm{TV}(F_l^0)}{S(F_l^0)}$, $l = 1, \ldots, k$, $\gamma^0 = \sum_{l=1}^k \lambda_l^0$, $I^0 = \emptyset$, $L = \emptyset$, $p = 0$

2: **Output:** partition $(C_1^*, \ldots, C_k^*)$

3: **repeat**

4:      $(F^{t+1}, \delta^{+,\,t+1}, \delta^{-,\,t+1})$ be the optimal solution of the inner problem (8)

5:      $\lambda_l^{t+1} = \frac{\mathrm{TV}(F_l^{t+1})}{S(F_l^{t+1})}$, $l = 1, \ldots, k$, $\gamma^{t+1} = \sum_{l=1}^k \lambda_l^{t+1}$,

6:      $\chi^{t+1} = \sum_{l=1}^k \frac{\mathrm{cut}(C_l^{t+1}, \overline{C_l^{t+1}})}{\hat{S}(C_l^{t+1})}$, where $(C_1^{t+1}, \ldots, C_k^{t+1})$ is obtained from $F^{t+1}$ via rounding

7:      **if** $\chi^{t+1} < \chi^t$ and $(C_1^{t+1}, \ldots, C_k^{t+1})$ is a $k$-partition **then**

8:          $(C_1^*, \ldots, C_k^*) = (C_1^{t+1}, \ldots, C_k^{t+1})$

9:          compute new ordering $\mathcal{O}_l, \forall l = 1, \ldots, k$ for $(C_1^*, \ldots, C_k^*)$ according to (10)

10:          $I^{t+1} = \bigcup_{l=1}^k \mathcal{O}_l^p$, where $\mathcal{O}_l^p$ denotes $p$ top-ranked vertices in $\mathcal{O}_l$

11:          $L = \{(i, j_i) \mid i \in I^{t+1}, \ j_i = \arg\max_j F_{ij}^{t+1}\}$

12:      **else**

13:          $p = \max\{2 |I^t|, 1\}$ (double the number of membership constraints)

14:          $I^{t+1} = \bigcup_{l=1}^k \mathcal{O}_l^p$, where $\mathcal{O}_l^p$ denotes $p$ top-ranked vertices in $\mathcal{O}_l$

15:          $L = \{(i, j_i) \mid i \in I^{t+1}, \ j_i = \arg\max_j F_{ij}^t\}$

16:          $F^{t+1} = F^t$, $F_{ij}^{t+1} = 0$, $\forall i \in I^{t+1}$, $\forall j \in \{1, \ldots, k\}$, $F_{ij_i}^{t+1} = 1$, $\forall (i, j_i) \in L$

17:          $\lambda_l^{t+1} = \frac{\mathrm{TV}(F_l^{t+1})}{S(F_l^{t+1})}$, $l = 1, \ldots, k$

18:      **end if**

19: **until** $\chi^{t+1} = \sum_{l=1}^k \lambda_l^{t+1}$ and $\gamma^{t+1} = \gamma^t$

---