[Reviews · NeurIPS 2014]

Submitted by Assigned_Reviewer_13

The paper presents a novel algorithm to solve the balanced k-cut problem. The k-cut criterion is defined by using the Lovasz extension of a set function. A relaxation of balanced k-cut functionals can be formulated using Lovasz extensions of cuts and of normalizing constraints. The appealing feature of this relaxation is the connection between convex Lovasz extensions and submodular set functions. The authors propose an algorithm that minimizes a sum of auxiliary variables that are lower-bounded by ratios. The minimization is further constraint by simplex and unique membership constraints.

Clarity: The paper is very well written and it appropriately refers to prior art.

Originality: There exists significant prior art and the novel idea modifies the optimization strategy. I consider the originality and in particular the novelty as limited but its significance due to the success in the experiments might justify publication of this paper at NIPS.

Significance: Due to the surprisingly clear superiority of the new method, this paper might show high impact in the future despite its high similarity with prior art.

Minor comments:

Figure 1 has five subfigures but only four are explained in the caption. What is the fifth subfigure about?

Summary: A novel continuous relaxation for balanced k-cuts with a descent algorithm is presented and
the experimental evaluation looks very promising.

Submitted by Assigned_Reviewer_26

This paper proposes a new tight continuous relaxation approach to the k-cut problem which avoids the greedy recursive splitting technique required for other approaches.

The generality of the monotonic descent method make it of general interest.

While the method shows clear promise in terms of finding the k vertices sets, the issue of estimating k itself is not really addressed.

Quality: The conclusions of the paper seemed fair based on the detail written. A conclusions section with caveats and future work would have improved the quality however.

Clarity: The majority was well-written, if a little dense at times. Missing a conclusions/discussion section.

Originality: The idea seemed original but I find it difficult to comment on this due to my lack of expertise in this area.

Significance: Difficult to say.

Small issues:

Move the "Balanced k-Cut" to the second line of the title

"Clustering" on line 17 should not be capitalised.

Line 42: "frequently outperform" rather than "outperform frequently"

Line 57: change "with small amount of label information" to "with a small amount of label information,"

Add letters (a), (b), etc to graphs in Figure 1.
Summary: An interesting theoretical idea but the paper is lacking in a conclusions section or discussion of caveats. The reframing of the cut problem in terms of the continuous relaxation proposed was of interest as was the monotonic descent method proposed for implementing its solution.

Submitted by Assigned_Reviewer_30

This paper proposes a new algorithm for the balanced k-cut objective function using continuous relaxation, for both the classical clustering setting with no labeled data, as well as the transductive setting. Thorough experimental results compare the new method against many other variation of spectral clustering, and demonstrate that the new method often outperforms previous ones. Particularly, the new algorithm consistently finds better balanced k-cuts, but also, somewhat surprisingly, it is able to find better partitions based on other related objective functions, even against algorithms that are designed for those functions.

Minor comments:
- Page 1, at end of the first paragraph of the introduction, switch the order of the words “outperform” and “frequently.”
- Please alphabetize the bibliography.

Summary: A continuous relaxation of balanced k-cut is presented. Thorough experimental results show notable qualitative improved other previous related techniques.
Author Feedback
Author rebuttal: We thank all the reviewers for their comments and will integrate all the suggestions in the final version.

We would like to emphasize the originality and the significance of our paper.

Originality:
------------

Ratio and normalized cuts are heavily used in graph-based clustering and image segmentation. Exact continuous relaxations for these cut criteria have been recently established for the bi-partitioning case (no. of clusters, k = 2) [9,10,11] and have been shown to be superior in performance to standard spectral clustering. However, for the multi-partitioning case (k > 2), [9,10,11] had to resort to a suboptimal greedy recursive splitting.

The first direct minimization method for the multi-cut criterion has been proposed in a recent paper [13]. However, as pointed out in Section 3 of our paper, their formulation has a serious flaw: their method can be shown to converge to a trivial solution (see Corollary 1 and Figure 1 of Section 3) which does not yield a feasible k-way partition. The authors of [13] were generous to provide us with their code; in order not to converge to a trivial solution, what they do in the code is early stopping. In contrast, our approach has a well-founded theoretical basis and does not require such ``tricks''. Moreover, their algorithmic scheme for the continuous formulation has no descent guarantees whereas we prove monotonic descent for our method. Owing to these two factors, it is not surprising to us that our method performs so well (as reviewer 1 and 3 mention), rather we found it surprising that their method, with their early stopping trick, can sometimes give reasonable results.

The main *contributions* of our paper are:

(i) a novel continuous relaxation for the general balanced k-cut problem (which includes ratio/normalized cut problems as special cases) and a guarantee that rounding the solution of our continuous relaxation yields a k-way partition. This is in strong contrast to [13]; we show that their method, in most cases, converges to a trivial solution (see Corollary 1, Section 3 for the exact statement).

(ii) an algorithmic framework with a monotonic descent guarantee for solving the difficult sum-of-ratios problem (a non-convex, non-smooth problem) - the sum-of-ratios problem is considered to be the hardest ratio problem within the optimization community.

(iii) the best solver for the balanced k-cut problem as shown by the quantitative results (Section 5) (note that we achieve the best cut in 81-99% of the cases among all methods and our method is strictly better than all others in 45-62% of the problems).

Impact/Significance:
--------------------
We will add a conclusion section in the final version of the paper as suggested by reviewer 2. We believe that this paper will have high impact because of three reasons:

1) Spectral clustering, which is based on ratio/normalized cut, is a well established graph-based clustering method. As shown in our quantitative experiments, our method provides by a large margin the best cuts among all competing methods - this will also lead to improvements in other applications such as image segmentation.

2) Graph-based methods: our generic framework enables one to directly minimize not only the established criteria such as ratio/normalized cut but also new application-specific balancing functions. Moreover, the integration of prior information in terms of additional constraints such as must-link or cannot-link constraints is, in our formulation, relatively straightforward.

3) Algorithm: We have proposed an algorithm which guarantees monotonic descent for the difficult sum-of-ratios problem (non-smooth, non-convex). As ratio problems often appear in dimensionality reduction and other unsupervised learning problems, we believe that our algorithm is of high interest in other applications as well.

Individual answers:
--------------------
Reviewer 1: "What is the fifth subfigure about?"

Sorry, this is a typo - there is two times d) - the last d) should be e) and it should be "In b)-e) we color ..."

Reviewer 3: "Particularly, the new algorithm consistently finds better balanced k-cuts, but also, somewhat surprisingly, it is able to find better partitions based on other related objective functions, even against algorithms that are designed for those functions."

Please note that in our general framework we can minimize all the different cut criteria directly which just amounts to changing the balancing function which, in the algorithm, corresponds to a simple change of the choice of the subgradient s^t_l in (9).